# Reward foraging task and model-based analysis reveal how fruit flies learn value of available options

Sophie E. Seidenbecher[1,2], Joshua I. Sanders[1,2¤], Anne C. von Philipsborn[1,2], Duda Kvitsiani[1,2]*

**1** Danish Research Institute of Translational Neuroscience - DANDRITE, Nordic-EMBL Partnership for Molecular Medicine, Aarhus, Denmark, **2** Department of Molecular Biology and Genetics, Aarhus University, Aarhus, Denmark

¤ Current address: Sanworks LLC, Stony Brook, New York, United States of America
* kvitsi@dandrite.au.dk

**Data Availability Statement:** The experimental data are deposited on figshare with DOI: 10.6084/m9.figshare.12682067, url: https://figshare.com/s/

## Abstract

Foraging animals have to evaluate, compare and select food patches in order to increase their fitness. Understanding what drives foraging decisions requires careful manipulation of the value of alternative options while monitoring animals choices. Value-based decision-making tasks in combination with formal learning models have provided both an experimental and theoretical framework to study foraging decisions in lab settings. While these approaches were successfully used in the past to understand what drives choices in mammals, very little work has been done on fruit flies. This is despite the fact that fruit flies have served as model organism for many complex behavioural paradigms. To fill this gap we developed a single-animal, trial-based decision making task, where freely walking flies experienced optogenetic sugar-receptor neuron stimulation. We controlled the value of available options by manipulating the probabilities of optogenetic stimulation. We show that flies integrate reward history of chosen options and forget value of unchosen options. We further discover that flies assign higher values to rewards experienced early in the behavioural session, consistent with formal reinforcement learning models. Finally, we also show that the probabilistic rewards affect walking trajectories of flies, suggesting that accumulated value is controlling the navigation vector of flies in a graded fashion. These findings establish the fruit fly as a model organism to explore the genetic and circuit basis of reward foraging decisions.

## Introduction

All food foraging animals have to learn the value of different food patches (harvesting potential) in order to better choose better among available options [1–3]. Making foraging decisions is complicated by environmental uncertainty, for example by variation over time in quantity and quality of food patches [4]. How do animals learn the value of different options in uncertain environments and what exact strategies and rules do they follow?

2a5f8c5a2ff0dd16758e. The Matlab scripts for behavioral control: 10.6084/m9.figshare. 12681509, url: https://figshare.com/s/ 7d5edd2801ae07df3c28.

**Funding:** DK, SS, JS, AP were funded for this work by Lundbeck foundation grant DANDRITE-R248-2016-2518. Website of the foundation is https:// www.lundbeckfonden.com/en/ The funders had no role in study design, data collection and analysis, decision to publish, or preparation of the manuscript.

**Competing interests:** No authors have competing interests.

In laboratory settings value-based decision-making tasks approximate behaviour of food foraging animals by parametrically manipulating the value of available options [5]. These tasks in combination with formal learning models [6–8] revealed decision variables that guide choices. In mammalian species this framework provided us with understanding of the algorithmic basis of value-based decisions and its neural implementation mechanisms [9–11]. However, due to the lack of the same approaches in the fruit fly foraging field, it is an open question how flies accumulate value of options and how formal learning models capture the value accumulation process.

The reinforcement learning (RL) framework provides a simple set of rules that enable agents to learn the value of options by trial and error. Agents that follow the RL algorithm can estimate the average rate of rewards (expectations) by integrating past rewards over multiple past trials [6]. In most of the RL models this integration is leaky, meaning that immediate past rewards have stronger effects on choices than rewards further in the past [6]. Not surprisingly, the RL framework has been successfully used to explain animal behaviour in many learning paradigms [12–14]. Besides its utility to explain animal behaviour, the RL framework is used to extract decision variables that are not directly observable to the experimenter, such as the value that animals assign to available options [5]. Furthermore, using model comparison [15] one can select the best predictive and generative RL model and see what behavioural strategies are used by animals. For example, according to standard Rescorla-Wagner (RW) RL models [12] the unchosen option values (options that are not selected by the agent on a current trial) are "frozen" and updated only after the animal samples that option. Alternative RL models assume [16] that unchosen option values decay (the animal forgets) until the animal chooses that option again. Forgetting of unchosen option values can be advantageous in highly dynamic environment where storing old associations in memory can be both costly and non-adaptive. Indeed theoretical and experimental evidence suggest that flies use the latter strategy to learn new associations [17–19]. However, direct predictive and generative tests that would compare several RL models against each other are missing.

In order to understand how flies learn option values and how these values impact decision making and overall behaviors we designed a single-animal, trial-based probabilistic reward foraging assay. Probabilistic reward in the form of optogenetic stimulation of sugar receptor neurons [20] allowed us to parametrically modulate the value of available options. Tracking walking trajectories of these animals revealed how learned values affected choices and overall walking kinematics of flies.

We found that fruit flies learn the value of options by integrating reward history. We also discovered that fly behavior is consistent with the RL model that accumulates chosen option values and forgets unchosen option values. The RL model explained the somewhat counterintuitive responses of flies to first experienced rewards. We observed that the value of first rewards varied as a function of time, meaning that flies choose to return more often to first rewards that occurred early in the session compared to first rewards that occurred late the session. Finally, we observe that flies' walking angles became narrow as a function of experienced reward probabilities, suggesting that these animals update their navigation vectors in proportion to learned values.

## Materials and methods

Single *Drosophila melanogaster* males were food restricted and placed in a linear track arena, see Fig 1A, which they were free to explore. The trehalose sugar-receptor neurons *Gr5a* [20] were chosen to express the light-activated ion channel Channelrhodopsin Chrimson [21], by means of the *LexA-LexAop* system. The optogenetic fly foraging setup consists of a 3D-printed

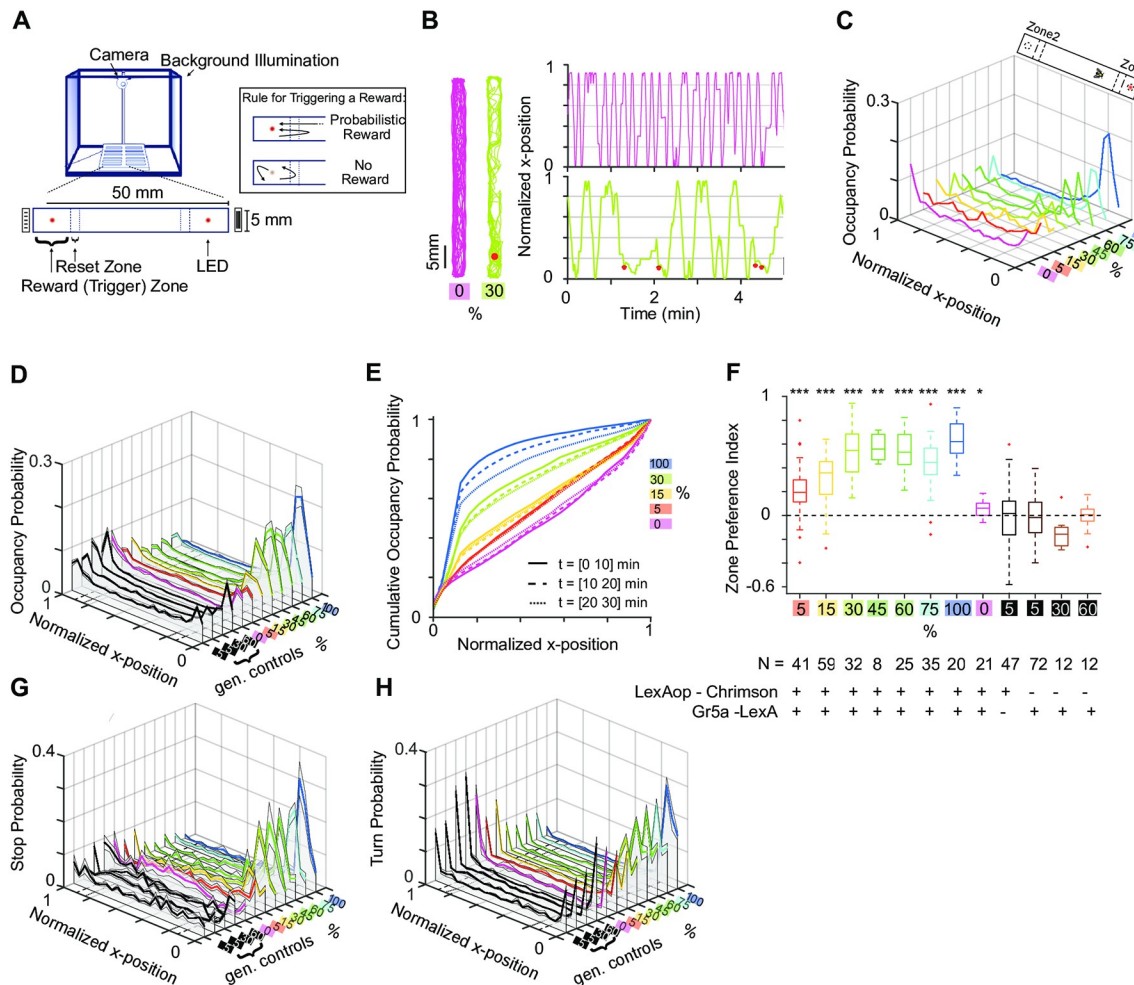

**Fig 1. Place preference as a function of *Gr5a*-receptor-neuron stimulation.** For optogenetic stimulation we used flies that express Chrimson in *Gr5a* neurons. As a genetic control we used *LexAop-Chrimson*/+ and *Gr5a-LexA*/+ flies. gen.controls for first 5% refers to *LexAop-Chrimson*/+ and 5%, 30%, 60% to *Gr5a-LexA*/+ genetic lines. **A** Single-fly optogenetic foraging set-up. A system of 12 linear track arenas is placed in a behavior box with uniform white background illumination and monitored by a webcam from above. Each arena contains two stimulating LEDs (= 624 nm) mounted below each of the track. Reset and trigger zones (short and long dash) are not visible to the flies. Distal visual cues were drawn behind each trigger zone: black and white stripe patterns with different orientations on each side. Inset: Rule for triggering a probabilistic flash of light. A flash is triggered only when the fly enters the reset and the reward zones in that order. **B** Left: Two dimensional walking traces of an unstimulated example fly (magenta) and 30% probability stimulated example fly (green). Light stimulation was delivered to only one side of the arena, marked with a red dot. Right: One dimensional walking trace over time of the same example fly. Stimulation events are marked with red dots. **C** Occupancy distribution of example individual from 0-100% stimulation conditions. **D** Occupancy distribution of fly populations that experienced the same stimulation probabilities as in **C**. Solid lines: Mean, shaded regions around mean: ± SEM. Genetic controls used as in **F**, *LexAop-Chrimson*/+ and *Gr5a-LexA*/+ do not express Chrimson. **E** Cumulative occupancy distribution over 10 minute intervals across time. Higher stimulation probability leads to a decrease of zone preference over time. **F** Zone preference index of stimulated fly populations and genetic controls (*LexAop-Chrimson*/+ and *Gr5a-LexA*/+) are shown as box plots (median, quartiles, entire range and outliers). Equal zone preference at preference index value 0 marked with black dashed line. Positive values indicate preference for zone 1 (stimulated) and negative values for zone 2. Stimulated flies have a significant zone preference of the stimulated zone over the unstimulated zone. (*: p < 0.05; **: p < 0.01; ***: p < 0.001, Kruskal-Wallis test with multiple comparisons). **G** Stop distribution in the arena for the fly populations. Solid lines: Mean, shaded regions around mean: ± SEM. *LexAop-Chrimson*/+ and *Gr5a-LexA*/+ were used as genetic controls. For definitions of stops and turns refer to materials and methods **H** Turning distribution for the fly populations. Solid lines: Mean, shaded regions around mean: ± SEM. *LexAop-Chrimson*/+ and *Gr5a-LexA*/+ were used as genetic controls.

platform with 12 identical linear arenas of 5 by 50 mm, each for a single fly, similar to Ref. [22]. The arenas were each separated by black barriers to block the spread of light to neighbouring arenas. Red light LEDs ($\lambda = 624(631)$ nm, Vishay VLCS5830) were mounted vertically below the arena floor to illuminate each of the arenas' two trigger zones. LED brightness and flash timing were controlled in real-time by a pair of microcontroller boards (Arduino Due), which provided PC control over the arena LEDs. The setup was surrounded on three sides by acrylic panels (EndLighten, Acrylite), each lit by a strip of white LEDs mounted along the end to provide white uniform background illumination and a water reservoir for humidity. The setup was monitored from above with a webcam (LifeCam Studio, Microsoft), fitted with a short-pass filter (FESH0600 Thorlabs) to block red light from the stimulating LEDs. Centroid fly-tracking and stimulation triggering were controlled by custom written MATLAB (Mathworks) scripts. Stimulation flash timing was controlled by custom firmware for Arduino Due, written in C++/Arduino language. In the camera view at the ends of each arena additional ROIs are defined to separate "reward" and "reset" zones. Using two zones allowed us to avoid self-stimulation when the fly simply stayed in the rewarded location. The reward and reset zones extend 6 mm and 3 mm, respectively, and zones of the same type are of the same size. Probabilistic rewards are triggered when the fly crosses the reset zone and enters the reward zone, in that order. Refer to the inset of Fig 1A for a depiction of the trigger rule. The stimulation duration was 0.05 seconds.

## Fly strains and rearing

Flies were housed under a 12 h:12 h light:dark cycle at 25˚C and 60–70% humidity on cornmeal, oatmeal, yeast and sucrose food. For all experiments 3-6 day old males were used, which were starved for 10-12 h prior to testing, while supplying water via a wet cloth. Flies were then transferred to the arena using an aspirator and left in the arena for 2-10 hours. The following strains were employed: *Gr5a-LexA* (gift from Kristin Scott [23]), *LexAop-Chrimson* ([21], w1118; P {13XLexAop2-IVS-CsChrimson.mVenus} attP40, Bloomington 55138), *Canton S* (from A.v. Philipsborn). The flies expressing *Chrimson* were fed all-trans retinal (ATR, Sigma Aldrich, CAS Number: 116-31-4) for 2-3 days before the starvation period. ATR food was prepared by mixing normal food with ATR to reach a 400 µM solution and then covered with aluminum foil to avoid degradation. Flies fed on ATR food were kept in the dark under aluminum foil cover.

## Experimental conditions

*Chrimson* >*Gr5a* flies were tested in eight different single-sided stimulation conditions; with 0, 5, 15, 30, 45, 60, 75 and 100% stimulation probability. In a second series of experiments, flies were tested under double-sided stimulation conditions, with 5-5% and 15-15% stimulation probability.

## Post-processing of walking data

Walking traces were cleaned of missing data points and jumps in the centroid contrast tracking and filtered with a butterworth filter using a cutoff frequency determined from camera jitter. Next, a trial structure was defined and data from flies with less than 50 trials was excluded from further analysis.

## Definition of observables

Stops were defined by speeds below a value of $|v| \leq 0.01$mm/s, which is governed by the resolution of our tracking system and corresponds to a movement of less than one pixel between

two timestamps. Turns were defined by velocity sign changes since our setup is effectively one-dimensional.

## Logistic regression

Regression analysis was performed on return choices against their reward history for individual flies and fly populations by averaging over individuals from the same experimental condition. Due to the binary output variable we used logistic regression. Here a weighted sum of the input vector $X_i$, $i \in \{1, \ldots, M\}$ that consists of two elements for each trial back i of past rewards and return choices is assumed to be a logit function of the dependent binary output variable $y$ (return choice). To estimate the weights $\beta_i$ for each element of the reward and choice history, the weighted sum $h(x)$ is computed,

$$h(x) = \beta_0 + \Sigma_{i=1}^{M} \beta_i X_i \tag{1}$$

and used to define

$$y' = h(x) + \epsilon \tag{2}$$

where $\epsilon$ is the remaining difference (error) between $y'$ and the estimate of $y'$, $h$. $y'$ is a continuous latent variable that needs to be mapped to the binary output variable $y$. Thus, the probability of seeing $y = 1$ is a logistic function of $h$,

$$P(y = 1) = \frac{1}{1 + e^{-h(x)}} \tag{3}$$

Logistic regression yields estimates of the parameters $\beta_i$ from the data which can be used to make predictions.

To understand the values of the regression weights and what can be concluded about the fly behavior from them, we generated 100000 element reward vectors with different reward probabilities (5-30%). Under the assumption that the regression weights are determined by how often the flies returned to a stimulation and neglecting any reward correlations, we generated corresponding return choice vectors. The percentage of return choices following a reward was set to approximate "medium responsiveness", with 50% correspondence. There were no choices on unrewarded trials. The regression weights can be seen in S3 Fig. Logistic regression was implemented using Matlab function glmfit.

## Reinforcement learning models

The following reinforcement learning models [6] were applied to the data to identify potential underlying algorithms: a Rescorla Wagner (RW) model [12], a forgetting model where learning and forgetting rates are equal (termed FQ model) and a forgetting model where learning and forgetting happen at different rates (termed $FQ^{\alpha_F}$).

The RL models were fit to each individual fly using maximum likelihood estimation with the following log likelihood function

$$L = \frac{1}{N} \sum_{t=1}^{N} ((1 - c(t)) \cdot \log(1 - P(c(t) = 1)) + c(t) \cdot \log P(c(t) = 1)) \tag{4}$$

$c(t) = 1$ corresponds to a return choice and $c(t) = 0$ corresponds to no choice on trial $t$. The simple RW model has three parameters, $\alpha$, $\beta$ and *bias*, where $\alpha$ is the learning rate, determining the impact of the reward-prediction error, $R(t) - Q(t - 1)$, on the value update, where $R(t)$ is the reward at trial $t$ and $Q(t)$ is the value corresponding to a choice. $\beta$ is the weighting factor

of the value in the choice probability,

$$P(c(t) = 1) = \frac{1}{1 + e^{\beta(bias - Q(t))}}$$ (5)

A *bias* parameter was included, to account for the fact that the baseline return probability for a fly is below 50%. In this simple model, the value of a choice $c = 1$ is only updated, when the fly made a choice, and remains constant otherwise ($\phi = 0$ in Eq 6). To make the model slightly more realistic, a second RL model, the FQ model, was implemented, where the value of a choice was forgotten, if the fly didn't make a choice, with the same learning parameter $\phi = \alpha$ as in the value update equation. The third model had one additional parameter, a forgetting parameter $\phi = \alpha_F$, to allow for the more general case of different strengths of the learning and forgetting processes.

$$Q^{FQ}(t) = \begin{cases} Q^{FQ}(t-1) + \alpha(R(t) - Q^{FQ}(t-1)), & \text{if } c(t) = 1 \\ Q^{FQ}(t-1) - \phi Q^{FQ}(t-1), & \text{else} \end{cases}$$ (6)

Every fly was fit with 100 random initializations of these parameter sets for each model and the best parameters were selected by the corresponding highest log likelihood values, $\ln(L)$. Subsequently, the Akaike Information Criterion [24] (AIC) score was computed, to select the one that best fits the data, while taking the number of parameters into account.

To allow for predictive testing of the models, only half of every fly data was used to fit the parameter values and the other half was used to predict the flies' choices. The $F_1$ score was used as accuracy measure for every fly,

$$\text{precision} = \text{TP}/(\text{TP} + \text{FP})$$ (7)

$$\text{recall} = \text{TP}/(\text{TP} + \text{FN})$$ (8)

$$F_1 = 2 \frac{\text{precision} \cdot \text{recall}}{\text{precision} + \text{recall}}$$ (9)

with TP the rate of true positives (fraction of trials that the model predicted correctly fly's return choices), FP the rate of false positives (fraction of trials when model predicted returns but fly did not return) and FN the rate of false negatives (fraction of trials when model did not predict returns, but fly returned).

To test the models' generative power, 1000 sequences of 1000 trials each for the different experimental probability conditions were simulated.

## Results

### Place preference depends on optogenetic stimulation probability of sugar receptor neurons

Numerous studies show that the rewarding experience from a specific location in space changes behavior of animals so that they spend more time in that location, in other words, it induces place preference to that location [25–27]. In order to see how place preference changes as a function of experienced rewards we set up a single-fly, closed-loop optogenetic stimulation assay. Flies were food-restricted (10-12hr restriction) and received as a reward the sensation of sweet taste by the brief stimulation of trehalose *Gr5a* [20] sugar receptor neurons that expressed the Channelrhodopsin Chrimson [21]. In this assay a fly was free to walk in a linear track arena. We defined trigger and reset zones placed at each end of the arena to deliver light

pulses conditioned on the fly's behavior (Fig 1A). Single pulses of optogenetic stimulation with a fixed probability were delivered only when the fly crossed the reset and trigger zones as described in the inset in Fig 1A. Thus, simply staying in the trigger zone did not provide additional optogenetic stimulation. Each fly experienced only one stimulation probability condition and all flies were initially loaded into the center of the arena. Similar to previous studies [28–30], we observed clear effects of the light stimulation on behavior.

First, we looked at how kinematic variables evolved as a function of stimulation frequency. Since our setup can be seen as one-dimensional, we first focused our analysis on the walking traces along the x-axis (long axis). The optogenetic stimulation changed the walking patterns of the flies, from a more symmetrical coverage of the arena shown for one example fly in Fig 1B (magenta traces), to a stimulation zone localized occupancy (Fig 1B, green traces). By testing stimulation probabilities from 0—100%, we showed that place preference (measured by occupancy probability of x positions) positively correlates with the stimulation frequency, as shown for an individual fly (Fig 1C) and for the population (Fig 1D). At 5% stimulation probability, the population occupancy distribution (Fig 1D) was very similar to unstimulated and genetic controls, showing that a low stimulation probability is not sufficient to induce a significant place preference. We observed a temporal decay of the place preference (Fig 1E) with a probability dependent magnitude. This indicated a saturation or behavioural adaptation effect in response to the optical stimulation.

To quantify the flies' preference for the stimulation side we compared the flies' zone preference indices across different stimulation probabilities (Fig 1F). Preference indices were computed from the occupancy (time spent in one of the zones) distributions for each zone (within the reset zone boundaries), using

$$PI = \frac{Zone\ 1\ Occupancy - Zone\ 2\ Occupancy}{Zone\ 1\ Occupancy + Zone\ 2\ Occupancy} \qquad (10)$$

which produces preference index values between -1 and 1, meaning strong zone 2 or strong zone 1 preference, respectively, and indifference at PI = 0. There was a positive correlation of the stimulation probability and zone preference index. All stimulated genetic control animals (that did not express Chrimson) had preference indices around 0 and did not show difference ($p > 0.05$) in their preference for zone 1 vs zone 2, proving that this preference effect doesn't stem from simple attraction to the light. To further test if flies had developed the preference for one side of the arena during optical stimulation, we performed double-sided stimulation experiments. We observed similar levels of occupancy and place preference with these flies (S2A and S2B Fig).

These results suggest, that flies associate place with the rewarding experience of sugar receptor neuron stimulation.

## Optogenetic stimulation triggers local search behaviour

The optical stimulation may affect the place preference by stopping animals movement, by increasing local turning behaviour without linear displacement, or both. We looked at additional kinematic parameters to disambiguate these possibilities.

The optical stimulation decreased speed compared to non-stimulated trials (S1A Fig left and right panel) and increased the staying time duration of the fly around the stimulation zone (S1C Fig). However, we did not see any effect of optical stimulation on the average walking speed of the fly (S1B Fig). We neither saw a clear difference in speed distribution for walking paths toward the rewarded zone (in-walking path) vs away from the rewarded zone (out-walking path) (S1D Fig). Thus, the measured place preference could be due to an increase in the

frequency of stop events that happen upon optical stimulation. We defined stops as the speeds below a set threshold level, determined by the resolution of the camera (for precise definition of stops refer to the Materials and methods section). The frequency of stops increased with stimulation probability (Fig 1G). Increase in stop events was accompanied by an increase in turning events around the rewarded location (Fig 1H). Here we treated fly arena as effectively one-dimensional and therefore turns were defined as velocity sign changes that indicate reversal of walking direction. The probability of turns increased upon optical stimulation on the stimulated side, while it decreased for the unstimulated side (Fig 1H). The turn frequency of stimulated flies surpassed turn frequency of the control population at the arena walls (Fig 1H). We also looked at the temporal dynamics of turns on stimulated vs unstimulated trials. S3A Fig shows that turns persisted for longer periods in stimulated trials than in unstimulated trials. The same effect was seen in stimulated vs unstimulated flies (S3A Fig, rightmost panel). Therefore, we concluded that flies respond to optical stimulation by stopping and reversing their walking direction (turning). This behaviour is similar to local searches [31] previously described as a behavioural response to rewarding stimuli [29, 30] and suggested that the stimulation in our experiments was also rewarding.

Next, we asked if local searches exhibited adaptation to the reward probability. We analysed the polar plots of the angular distribution of the walking paths in the reset zone in S3C Fig. We have separated rewarded (solid lines) and unrewarded (dashed lines) trials. The search path of the rewarded fly had a larger variability in turn angles than an unrewarded fly's path, that only turns at the arena wall. The angular distribution of walking traces were significantly different for rewarded vs unrewarded flies, however we failed to see any significant difference on rewarded trials across different probability conditions (S3C Fig). The same was true for probability conditions of 5 and 15% (not shown here). This could be due to the fact that turns and stops on rewarded trials happen immediately following the optical stimulation and may elicit initial stereotypical behaviour. Therefore, we also looked at the unrewarded trials. The unrewarded angular distributions across conditions were significantly different from the unstimulated control flies (0%). These results suggested that local searches emerge upon reward encounter and they elicit probability independent responses.

## Flies return to the rewarded location as a function of reward probability, consistent with a value accumulation process

We reasoned that local searches signal active seeking of rewarded locations and asked if they resulted in animals returning to that location on a longer time scale. To measure this, we split the continuous walking trajectories into discrete trials. We defined a trial to be the time between two crossings of the same reset zone and the accompanying reward zone from the same direction, see Fig 2A. This means that within a trial, the fly visits a reward zone, makes the choice to either return to the same zone again without reaching the other zone (return decision), or to sample the other reward zone before returning. Trials also differed in whether or not the fly was rewarded when it entered the reward zone. In this way, we created a sequence of binary events consisting of a probabilistic reward followed by a binary choice to return or not. Fig 2B shows the return probability for all trials (rewarded and unrewarded) to each zone (one and two) for all tested probability conditions. In all stimulated conditions, there were significantly more returns to the rewarded zone than to the unrewarded zone, which was not the case for the unstimulated and genetic controls (Fig 2B).

The experienced reward rate and set reward probabilities differ due to the stochastic nature of the reward delivery. We also saw a positive correlation of returns with the experienced reward rate (Fig 2C), resulting in correlations between rewards and choices (S4C Fig) for

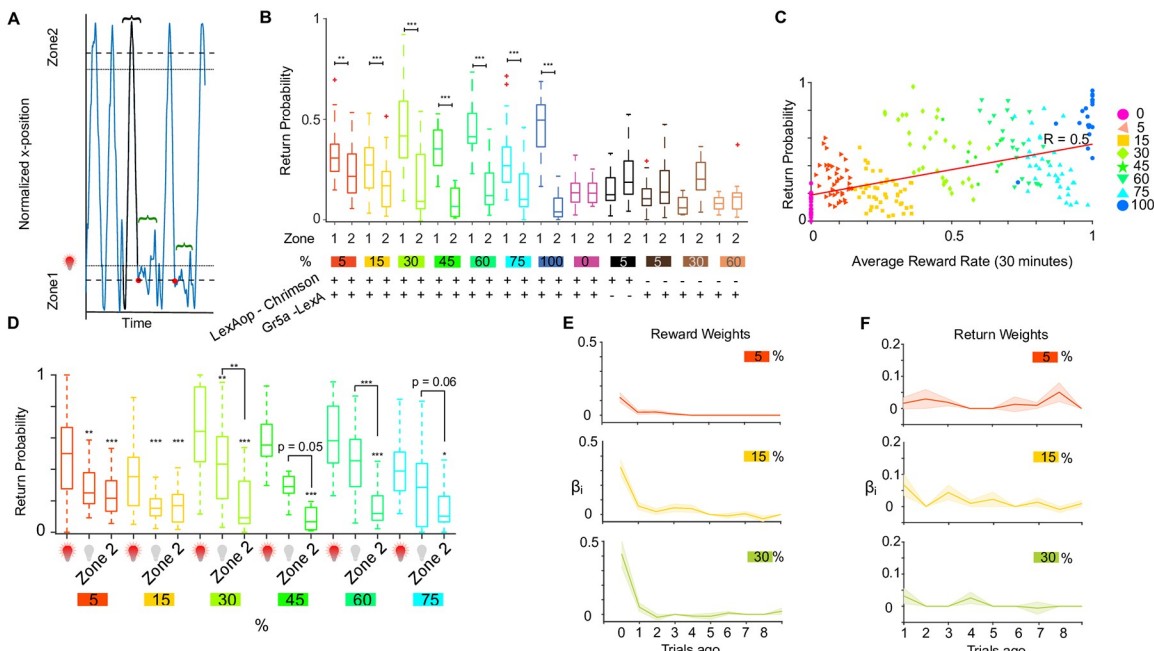

**Fig 2. Flies return to optogenetic stimulation site. A** Walking trace of one example fly over time in blue. Zone boundaries for reset zone are marked with dashed and for reward zone with dotted lines. Definition of a trial (highlighted trace in black) that consists of travel trajectory from entering the reset and reward zone, exiting the reset zone until entering the reset zone again in that order. Returns are defined as trials with walking trajectories that leave the reward and reset zone and return to the same reward zone, before reaching the other side's reset zone boundary. Black trace with the curly bracket represents a case for non return behaviour. Two examples of returns are shown in green curly brackets. **B** Total return behavior per probability condition to zones 1 and 2 (*: $p < 0.05$; **: $p < 0.01$; ***: $p < 0.001$, Mann Whitney U test) for stimulated flies and genetic controls (*LexAop-Chrimson*/+ and *Gr5a-LexA*/+) shown as box plots (median, quartiles, full range and outliers in red crosses are shown). Number of animals for each condition is the same as in (Fig 1F). **C** Return probability versus average reward probability within 30 minute behavioral session. Black line: Pearson correlation, $R = 0.5$, $p = 1e-16$ (Robust Correlation package by [32]). **D** Return behavior on rewarded trials (red light bulb), unrewarded trials (grey light bulb) and unrewarded zone 2. (*: $p < 0.05$; **: $p < 0.01$; ***: $p < 0.001$, Kruskal-Wallis test with multiple comparisons). Same number of animals were used as in (Fig 1F). **E** Logistic regression against the reward history. Solid lines: Population averages, shaded regions: ± SEM. Only significant coefficients ($p < 0.01$, computed using t-statistics) are shown. **F** Logistic regression of the return choices in the 5%, 15% and 30% condition against choice history for 10 trials into the past. Solid lines: Population average, shaded regions: ± SEM.

different probability conditions. This result suggests that flies accumulate values over rewarded trials as proposed by a previous study [29]. However, our optical stimulation protocol might sensitize or desensitize activated neurons to subsequent stimulations, thus masking the internal value accumulation process. Alternatively, fictive rewards in the form of optical stimulation may induce a phenomenon described as Caloric Frustration Memory (CFM). CFM is documented to reduce behavioral responses to the non-caloric artificial sweeteners [33]. In either case, we observed a reduction in behavioural responses over time in high frequency optical stimulation sessions. The behavioural reduction of responses to optical stimulation was most pronounced in the 100% stimulation case (S4A Fig) and was absent at lower probabilities both for single (S4A Fig) and double sided stimulation (S2G Fig). When present, these behavioural effects make it look as if animals show diminishing action values. To mitigate this problem most of our subsequent analysis was focused on performance of animals with low (up to 60%) stimulation probability conditions within a 30 min period from the start of the session. We observed that the return rate for a 30 min period drops by 35% (S4B Fig, upper panel), largely driven by flies that experience 100% stimulation condition (S4A Fig). The probabilistic reward delivery allowed us to further minimize the problem of behavioural reduction of responses to

high frequency optical stimulation and analyze how returns changed as a function of reward probability, on trials where animals had not been subjected to optical stimulation. Our data shows that as reward probability and reward rate increased, returns scaled up to the rewarded location on non-rewarded trials compared to the non-rewarded location (Fig 2D). Thus, we saw a strong evidence that despite behavioural reduction of responses to high frequency optical stimulation (S4B Fig) flies were accumulating an internal value of actions as a function of reward rates.

To look more directly into value accumulation we used logistic regression analysis to see how rewards in past trials contributed to current choices and computed the past reward effects (reward kernel) on current return choices [34]. We show that immediate rewards (0 trial ago rewards) had the strongest effect on current choices while rewards further back in the trial history had smaller contributions (Fig 2E). The same effects were seen with two-sided optical stimulation (S2E Fig, left and middle panels and S2F Fig). We limited this analysis to low probability stimulation conditions due to behavioural reduction of responses to high frequency optical stimulation as mentioned earlier. Based on our analysis we concluded that flies mostly rely on current rewards to make choices, but also incorporate rewards into their choices that happen further back in trials. A simulated fly that only responds to immediate rewards with 0.5 probability generates reward kernels unlike the ones we see in the animal data (S4D Fig). Namely, it has all zero regression coefficients for past rewards except for the immediate rewards. This is consistent with the idea that flies accumulate reward value over trials.

In some of the reward foraging studies using a probabilistic reinforcement structure [34, 35] not only rewards, but also past choices contribute to the animals' current choices. This is sometimes termed decision inertia [36]. To test if flies also exhibited decision inertia we regressed current choices on past choices. Please note that regression analysis was carried out using both past rewards and past choices using a generalized linear model as described in Eqs 1–3. This way we could identify independent contributions of past rewards and past choices to current choices (returns). Our analysis failed to detect any effects of past returns on current returns (Fig 2F, S2E Fig, right panel).

Overall, these results show that flies integrate reward history into choices.

## Reinforcement learning models that use forgetting and learning rates capture fly behaviour

Regression analysis of rewards and returns revealed that immediate rewards had strongest effects on choices. However, this analysis did not distinguish how flies update the value of chosen vs unchosen options (refer to the Methods section for details). Discounting or forgetting value of unchosen options would be advantageous for animals that forage in natural habitats where the potential number of feeding sites (options) is much higher than two. In such habitats keeping the memory of all visited options may compromise learning the new value of recently visited options. Therefore, flies, like mammals as documented in previous studies [37], may forget values of unchosen options. To see how unchosen option values are updated, we modeled the choice behavior within a reinforcement learning framework by comparing three RL models that use different update rules for unchosen options. One model freezes the value, one forgets the value with the same parameter as the learning rate $\alpha$ and the third forgets the value with a separate forgetting parameter $\alpha_F$ (Fig 3A). In order to compare the different RL models we used two different metrics. First, on each trial we computed the upcoming choices by each model (return or no return) given the reward and choice history that the fly experienced. We next computed the F1 score (see Materials and methods) that balances sensitivity (correct prediction of true positives) and specificity (correct rejection of true negatives) in assessing each

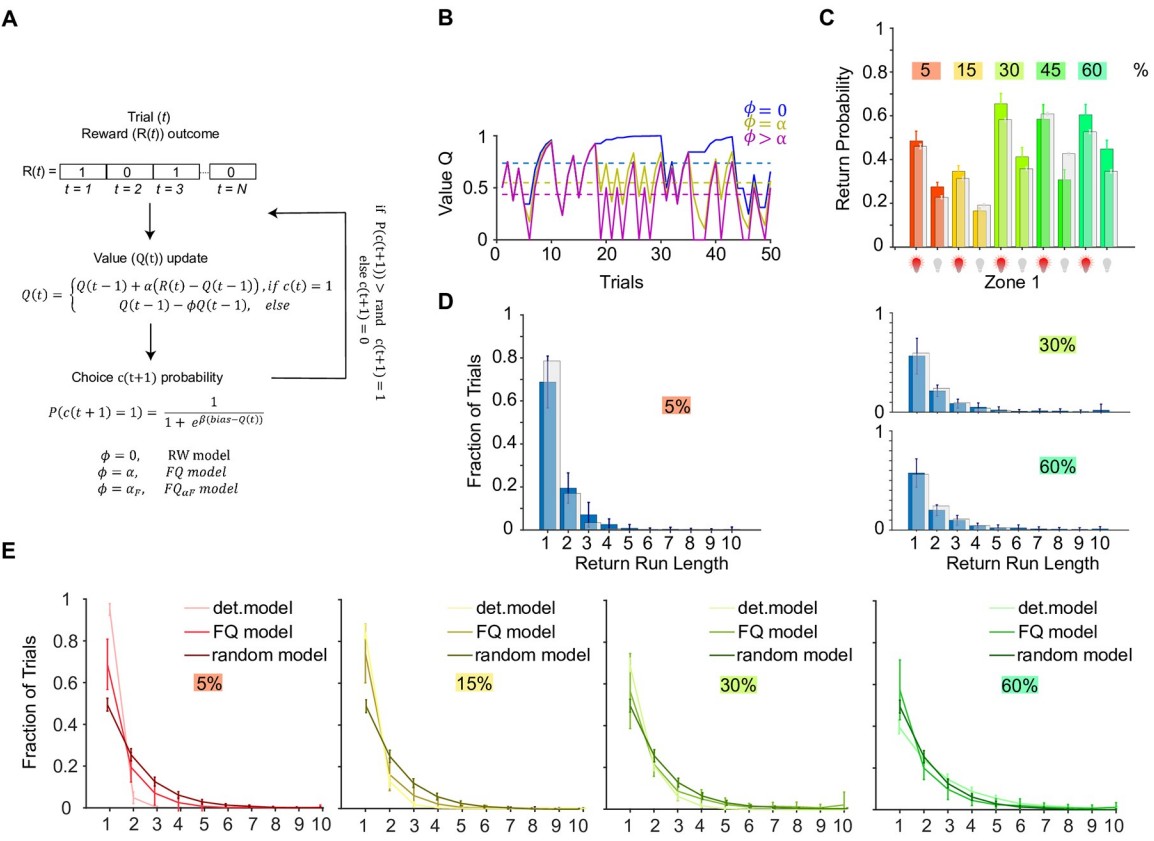

**Fig 3. Reinforcement learning model captures reward probability dependent returns. A** Schematics of RL algorithm that computes value (Q) and choice (same as return) probability (P(c)) for each trial (t) based on reward (R) history. Q value update on chosen options ($c(t) = 1$) is controlled by the learning rate α. If choice probability is above a random number (drawn from uniform probability distribution from 0 to 1) the agent returns to the rewarded location (choice equals 1) otherwise it does not. The different RL models differ in how they update the value of unchosen options, controlled by the parameter $\phi$, also illustrated in **B**. **B** Examples of value evolution over trials for each choice of $\phi$. RW model: blue, FQ model: yellow, $FQ_{\alpha_F}$ model: magenta. Dashed lines show average value for each model. **C** Generative test of the FQ model. Return probability separated by stimulated (red light bulb) and unstimulated (grey light bulb) trials to the rewarded zone per probability condition for the data (colored) and the model (grey). Bars: Population mean ± SEM. **D** Return choice run length histogram for population 5, 30, 60% fly data (blue) and the model (100 simulations, grey). Bars: Mean ± SEM. Run lengths are defined as consecutive returns to the same side. **E** Return choice run length generated by FQ model compared to deterministic (det) and random models for different reward probability conditions. The deterministic model generates returns with 0.5 probability when experiencing a reward, while the random model generates returns with probability of 0.5 independent of rewards. For each model Mean ± SEM are are shown.

model's predictive power. Second, we compared different RL models using generative tests. For this we first selected each model's parameters that best explained the observed data (Materials and methods). Next, using these parameters we used each model to generate choices in a simulated environment with two alternative options using the same reward probabilities as experienced by flies. We compared the number of consecutive return choices to the same option that each of our models and flies generated.

Examples of the evolution of the value over trials for all three models are depicted in Fig 3B. To account for the fact that the flies have a baseline return probability below 50%, we included a bias parameter. Model selection using the Akaike information criterion (AIC) score [24], a measure that describes how well a model fits the data by accounting for the number of parameters (S5A and S5B Fig), as well as predictive (S5C and S5D Fig) and generative tests (S5E and S5F Fig) slightly favored the second model which we termed forgetting-Q model, or FQ. The

best-fit parameter values show a high variability across flies and similar mean values across experimental conditions (S5B Fig). Predictive testing of the best-fit FQ model on the data yielded rather poor overall accuracy (S5C and S5D Fig). However, $F_1$ accuracy reached 80% when the model was fit on data that had roughly equal, or higher, numbers of returned to not-returned trials. Nevertheless, under generative testing the model was able to produce similar return probabilities as the flies (Fig 3C) and reproduced return run lengths (Fig 3D). The return run length was markedly different in FQ model that extracted parameters from fly behavior from a deterministic model that responded to rewards with 0.5 probability of returns and a random model that generated random returns independent of reward outcome (Fig 3E). We defined return run lengths as the number of consecutive returns to the same side. The same analysis performed on flies with two-sided stimulation also favored the FQ model (S2H Fig) that showed good predictive (S2J Fig) and generative performance (S2K Fig). However, we observed a much smaller spread of the FQ parameter values (S2I Fig). We think that this discrepancy stems from the data limitation for one-sided stimulation trials.

Overall our analysis favoured the FQ model as the more adequate description of fly behaviour.

## RL model explains time dependent nature of returns on first rewarded trials

In standard RL models the value of options starts from zero and is updated according to experienced reward outcomes (Eq 6) (Fig 1A). However, animals may not be completely neutral to environmental or internal cues in the very beginning of the trial and may instead be attracted or repulsed by the mere fact that these stimuli are novel [38]. For example, flies may show some initial attraction to the edges of the arena in the beginning of the assay. In this case the value update process would start from a positive initial value rather than from zero. To gain further insights into the value update process we looked how flies responded to first rewarded trials.

First we noticed that returns on the first rewarded trials were influenced by the reward probability (Fig 4A). One possible explanation could be that the flies were sensitive to the timing of the first reward in the session as mentioned earlier. To further elucidate behavioral mechanisms that drive this phenomenon we looked at the delay of the first rewards from the start of the session using both trial- (Fig 4B and 4E) and time-based analysis (Fig 4C and 4D). Both analyses showed a similar trend of decreased return probability with increased delay. We observed the same effect of the first rewarded trial on return probability when both sides were used to deliver optogenetic stimulation (S2D Fig). The arena on its own may have exerted a somewhat attractive effect on flies as control flies that never experienced rewards showed above baseline level of returns that decreased to baseline (Fig 4D and 4E, insets). This decay was fast as no change in returns were observed on first and later trials (Fig 4C, magenta circles connected with grey line) on a time-scale of minutes. The time course of returns could be also explained by an alternative hypothesis. Namely that flies when introduced to the arena at the beginning of the session have generally increased locomotion resulting in higher turning behavior that ultimately results in increased return behaviour. If so we should have seen the same effects on time evolution of turning behavior. We failed to detect any observable decay on time evolution of turning behavior on non rewarded trials (S3A Fig, lower panel).

Our previous analysis suggested that the value accumulates as the number/probability of rewarded trials increases (Fig 2D). If so, the timing of the first rewarded trial may have affected the subsequent return probability as the value should be higher for early vs later occurring first rewards. For this we looked at the return probability on all subsequent trials when first rewards

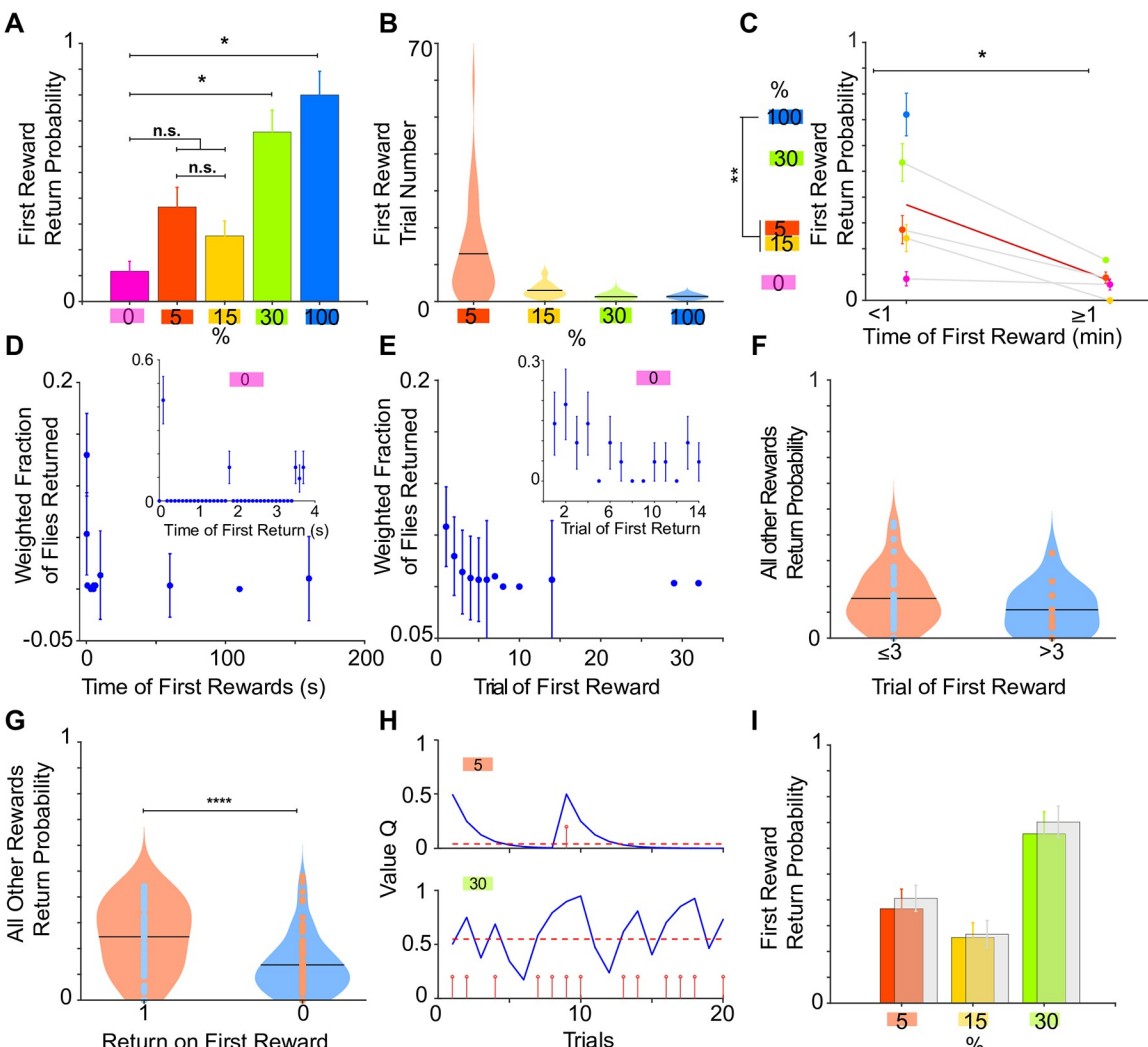

**Fig 4. The timing of first rewards affect return frequency. A** Return behavior upon first reward per probability condition. For unstimulated controls (0%, magenta) the return probability was computed for the first trial. (*: p < 0.05; pairwise Fishers exact test.) **B** First rewarded trial number per probability condition. Black lines: mean. **C** Return behavior upon first reward within the first minute of recording and after the first minute. Blue circles without a connecting grey line correspond to conditions where the first reward always happened within the first minute. Red line: average return probability. Magenta circles and dark grey line: unstimulated controls. Error bars: ± SEM. (*: p < 0.05; **: p < 0.01; ***: p < 0.001, Kruskal-Wallis test with multiple comparisons and Welch's t-test). **D** Fraction of flies (independent of stimulation probability) that returned to a first reward within the first 200s (time bins: 0.03s between 0 and 1s, 0.1s between 1 and 9s, 50s from 10 to 200s and 100s from 300 to 1000s). Inset: Fraction of unstimulated flies that returned for the first time since the session start against time (time bins of 0.1s). Error bars: ± SEM. **E** Fraction of flies that returned to a first reward within in the first 30 trials. Inset: Fraction of unstimulated flies that returned for the first time against the trial index. Error bars: ± SEM. **F** Return probability to all other rewarded trials when the first reward happened within the first 3 trials (orange violin) or after the first 3 trials (blue violin). **G** Return probability to all other rewarded trials depending on whether the fly returned upon the first reward (orange violin) or not (blue violin). (***: p < 0.001, Welch's t-test) **H** Two examples for the evolution of the RL value Q (blue curve) over trials. Upper figure: 5% stimulated fly. Lower figure: 30% stimulated fly. Red stems: stimulation events. Red dashed line: average value. **I** Transparent grey bars: RL model's prediction of first reward return rate. (Color bars as in **A**).

happened within the first 3 trials (this number was chosen since the return probability does not change if the fly is rewarded after the 3rd trial (Fig 4E)) or later (Fig 4F). We could not detect a significant difference in return probability between these groups, suggesting that except for the first few trials the behavior of the animal was not affected by the timing of the first rewards. There could be individual differences to novelty that may indicate the flies'

sensitivity to rewards in general. Therefore, we separated flies that showed return on first rewarded trial from flies that did not return on the first rewarded trial and looked at the return probability on all subsequent rewarded trials (Fig 4G). We show that return behavior on the first rewarded trial is a good predictor of future returns and may reflect individual differences among flies.

To formally account for the observed responses of flies on the first rewarded trials we incorporated this in our RL models and assumed that option values (in our case zone 1 and zone 2 of the arena) are not set to zero initially, but rather start with some default positive value that over time decays (Fig 4H). Note, that this simple model qualitatively explains return behavior in control flies that never experienced optical stimulation. We also tested if such RL model could correctly predict flies' returns to first rewards. The FQ RL model indeed generated similar return probabilities (Fig 4I).

The high return behaviour at the beginning of the session could have been due to the animal's motivation to escape the arena and not due to the attraction of flies to the novel environment. Because of this flies may have initiated more turns at the beginning of the session. We think this scenario is less likely, although we can not completely rule it out. If animals were in escape or aversive state they would have likely assigned less value to the first experienced rewards and thus showed less returns to those rewards than to the subsequent rewards. Our data showed opposite trends.

## The curvature of walking trajectories depends on accumulated value

The experienced reward rate or option values had a clear impact on return behavior in flies. However, due to the binarization of returns this analysis did not reveal how accumulated value affected walking trajectories of flies on a trial by trial basis. We therefore looked at the two-dimensional walking paths of flies as a function of accumulated value.

Previous work [39] has shown that inbound paths of flies to their feeding sites are more straight than outbound paths, suggesting that path integration mechanisms help animals reach their feeding sites using shorter routes. Here we used a similar approach and decided to look at how flies navigated towards and away from their rewarded location as a function of accumulated value. We looked at the angular distribution of walking paths on rewarded and non-rewarded trials as a function of reward probability. Angular distribution was calculated in the following way: For each individual walking $x$- and $y$-traces, we compute the angle between the fly's position at time $t$ $(x_t, y_t)$ and the next time step $t + 1$ $(x_{t+1}, y_{t+1})$, by computing the arctan between them. To ensure all angles are computed from the same reference vector, the angles are shifted by pi when the orientation of the trajectory changes. The more curved a path is, the more uniform the corresponding angular distribution gets, which translates into a higher angular distribution entropy (Fig 5A). First we noted that out-walking (walking away from the rewarded location) paths generally had slightly higher spread in angular distribution compared to in-walking paths (walking towards rewarded location) (Fig 5B and 5C). This difference did not reach statistical significance. We noted, however, a consistent trend in the reduction of the entropy in angular distribution of in-walking paths as a function of the reward probability (Fig 5D) ($p < 0.05$ for 5 and 15% reward probability compared with the 100% reward probability). Thus, flies choose to walk more straight paths towards the rewarded locations as a function of accumulated value.

## Discussion

We developed a single-fly, trial-based optogenetic reward foraging assay to study how parametric manipulation of value in the form of probabilistic rewards was changing the

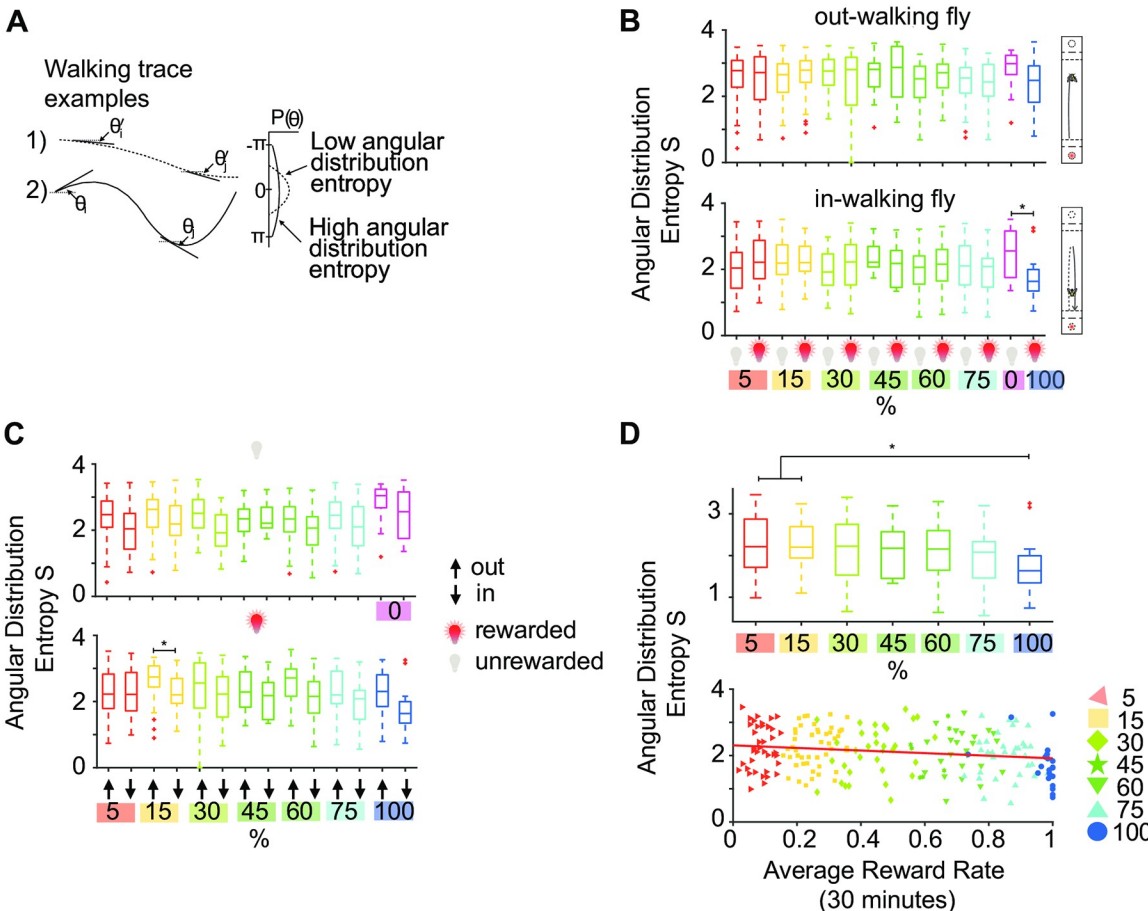

**Fig 5. The accumulated value affects walking direction of flies. A** Sketch of path angular distribution analysis. The "straighter" path 1) is characterized by a narrow angular distribution, while the more curved path 2) has a broader angular distribution. If the distribution is narrowly peaked, it has a smaller entropy S than a broader distribution. **B** Angular distribution entropy of in- and out-walking paths (box plots with median and quartiles shown), on rewarded (red light bulb) and unrewarded (grey light bulb) trials. For out-walking (away from the rewarded zone, measured from the reset zone) trajectories, there is no difference between rewarded and unrewarded trials. For in-walking trajectories (from the position of return to the reset zone), the rewarded trials have a smaller entropy, corresponding to more straight paths. This is significant for 100% compared to unstimulated controls (*: $p < 0.05$, Kruskal-Wallis test with multiple comparisons). **C** Same data (box plots with median and quartiles) as in **B** but sorted by unrewarded (top) and rewarded (bottom) trials. (*: $p < 0.05$, Kruskal-Wallis test with multiple comparisons). **D** There is a trend of more straight in-walking after a reward (indication of path integration) with increased reward probability. Top figure: only rewarded trials from **B** bottom (*: $p < 0.05$, Kruskal-Wallis test with multiple comparisons). Bottom figure: Same data plotted against each fly's experienced average reward rate. Red line: Pearson correlation, $R = -0.18$, $p = 0.005$ (Robust Correlation package by [32]).

foraging decisions of flies. The reward foraging task also allowed us to test formal models of learning. Using RL models we described how flies accumulate chosen option values and how they forget unchosen option values. Besides explaining decisions on a trial by trial basis the RL framework also accounted for the flies responses to the first rewarded trials and the direction of their walking paths.

Although our assay imitated some aspects of reward foraging behavior in natural habitats we note a number of limitations that accompany our studies. The linear arena largely limited the flies movements in one dimension (D). This may have masked some behavioural strategies that are only manifested in 2D. This is difficult to know with our current task design, but at least one study that used a 2D arena with a closed loop optogenetic stimulation arrived at similar conclusions [29]. We also considered the angle of walking direction as a function of

experienced reward probability and saw clear effects on behavior, thus suggesting that despite these limitations, 2D information of movement can be extracted. Another limitation of our studies is that optogenetic stimulation of sugar receptors without nutrient delivery may have attenuated the behavioral responses to such stimulation and thus resulted in a reduction of behavioral responses. There are reports that flies sense both sweet taste and nutritional value of sugar [40, 41]. Therefore, the quick reduction of returns on high probability stimulation sessions might be due to the lack of nutrient value of the optogenetic stimulation. Support for the role of nutrient value of ingested food is documented in flies, as artificial sweeteners reduce appetitive responses [33]. To remedy this problem we focused our analysis on relatively stable periods of performance. Finally, RL models are designed to study decision making tasks that use binary variables like choices, yet behaviours themselves operate on a continuous scale [42–45]. Therefore, one alternative would be to develop computational models that take into account continuous variables instead of binary ones [46, 47]. These models should better predict animal behaviour, but this may come at the expense of model complexity. Here we opted for simpler and easy to interpret computational models that have been heavily used in animal decision making field.

Despite these limitations our study made discoveries that can be also generalized beyond the species studied. In reward foraging decision making tasks value is manipulated by controlling the magnitude, probability or timing of rewards. The probabilistic optogenetic stimulation offered a better way to control value of rewards than manipulating stimulus duration or intensity. This is due to the fact that it is unknown how optical stimulation intensity or duration translates into reward value. Surprisingly, in the fruit fly behavioural field probabilistic reward foraging decision making tasks are missing. Although pavlovian and operant tasks have been extensively used in the fruit fly learning field [48–50], these studies do not typically control the value on a continuous scale. Different from learning paradigms, the manipulation of value is necessary to understand what decision variables drive animals' choices on a trial by trial basis. Our analysis revealed that reward history integration drives animals' choices as observed in other species [11, 34]. We also note that choice history had no observable effects on future choices unlike what has been observed in rodents, monkeys and humans [11, 34–36]. These may indicate differences in how the mammalian and insect brain solves the same foraging problems. Choice history integration may help foraging animals to maximize rewards in habitats with diminishing returns [51], when energy reserves undergo depletion once animals start to consume it. It is plausible that the depletion rate of visited food patches is insignificant compared to unvisited patches in fruit fly natural habitats. Therefore flies may only integrate reward history and ignore choice history in their decisions. Testing the formal RL models revealed the commonality of decision making rules across different species. Namely it has been noted that RL models that forget unchosen option values are better predicting rodent and monkey behavior than standard RL models that "freeze" unchosen option values [52, 53]. This form of learning might free up memory from storing values of unchosen options and instead tune it to dynamic environment where only the most recent experience is retained in memory. Forgetting the value of unchosen options might be a passive or active process. Our model-based approach revealed that learning and forgetting use the same rate parameter ($\alpha$) to update the Q values (Eq 6). This suggests that forgetting is not a passive process and like learning is actively controlled by dedicated genetic and neural circuit elements [54, 55]. Interestingly the forgetting process is controlled in flies by the Dopaminergic neurons, the same class of neurons that play the key role in learning [56].

Overall, our reward foraging assay lays out the foundation for linking specific molecular players and neural circuits to decision variables captured by formal learning models using unbiased genetic screens or targeted manipulations of candidate genes and neural circuits.

## Supporting information

**S1 Fig. Population speed and characterization of trials. A** Speed distribution of fly populations. Left: stimulated trials (grey light bulb). Right: unstimulated trials (red light bulb). **B** Average walking speed per condition. **C** Trial length distribution of 5,15,30 and 100% condition populations. Solid lines show rewarded trials (longer) and dashed lines show unrewarded trials (shorter). **D** Speed distribution after a reward. Solid lines: out of the reward zone walking speed. Dashed lines: in-walking speed when returning from walking out (same trial as out-walking speed). In-walking is on average slower than out-walking. **E** Pictogram of out-walking and in-walking traces.
(PDF)

**S2 Fig. Experiments with double-sided stimulation. A** Occupancy distribution for 5% and 15% double-sided stimulation data. **B** Preference index. **C** Return probability to zones 1 and 2 for both conditions. **D** Left: Return behavior on the first reward (to either zone) compared to the first trial return for unstimulated controls. Right: First rewarded trial number. **E** Left and center: Logistic regression weights of returns against the reward history for both population data to each zone independently. Right: Logistic regression weights for returns against return choice history. **F** Pearson correlation for rewards (R) and returns (C). **G** Return behavior as 5-trial moving average. Red curves: rewarded trials, blue curves: unrewarded trials to the same zone. **H** AIC score for the three RL models. **I** Best-fit parameter values of the FQ model. **J** Generative testing of the FQ model: comparison of the return probability (exp. data: colored, model: grey). **K** Generative testing of the FQ model: Return run lengths (exp. data: blue, model: grey).
(PDF)

**S3 Fig. Local search analysis. A** Top row: Turns, as a proxy for local search, (binned) in time since trial start for 30, 60 and 100/0% conditions. Middle row: histogram of temporal turn distribution. Solid drawn curve corresponds to rewarded trials (top row) and dashed curve corresponds to unstimulated trials (bottom row). Bottom row: turns in time since trial start for unstimulated trials. **B** Top row: returns on rewarded trials in time since trial start for the same fly populations. Middle row: histogram of returns. Solid curve: stimulated returns, dashed curve: unstimulated returns. Bottom row: Unstimulated returns. **C** Polar plots of angular distributions of walking traces in the reset zone, for 30%, 60%, 100% populations and unstimulated controls (clockwise). Solid lines: rewarded trials, dashed lines: unrewarded trials. (****: $p < 0.0001$, two-way Kolmogorov-Smirnoff test.) **D** Comparison of return location (maximum position of a trial) for 5 and 15% single and double sided condition fly populations. Double sided cases have rewards in both zones and thus returns to both zones are separated.
(PDF)

**S4 Fig. Stability of return behavior over trials and reward-choice correlations. A** 5-trial moving averages of the returns over trials for 0-100% stimulation probability conditions. Red curves show returns upon rewards, blue curves show returns to the rewarded zone without rewards and green curves show returns to the unstimulated zone.**B** Upper panel, return probability as a function of elapsed time from the start of the behavioral session. Single exponential fit (in red) to observer data. Blue circles indicate return probability of flies averaged across all reward probability conditions across time. At 30 min from the start of the behavioral session average probability of return drops by 35% (black intercept). Lower panel shows return probability averaged across all probability conditions as a function of number of rewards experienced by flies. Number of rewards and corresponding return probability are shown in legend **C** Pearson correlation of rewards and returns (choices) for 5,15 and 30% conditions. **C** Logistic

regression of simulated data to rewards. Simulated data was generated with 50% return probability upon a reward. Curves show regression weights for different stimulation probabilities (5-30%). **D** The regression weights for experimentally observed data (color coded for each probability condition) from flies run on different probability conditions, deterministic model (blue curve) that responds to only immediate rewards with 0.5 return rate and difference between them (magenta bars).
(PDF)

**S5 Fig. Reinforcement learning model selection, predictive test and generative test. A** AIC scores for the three RL models on 5-60% data. The lower the AIC score, the better the model captures the data while excessive parameters are punished. **B** Best-fit parameter values of the FQ model for each fly (circles) and population averages (solid lines in the violins). **C** Predictive test of the FQ model. Number of flies that could be predicted with more than 50% accuracy ($F_1$ score) for each model. Total number of flies per condition: $N^{5\%} = 94$, $N^{15\%} = 70$, $N^{30\%} = 56$, $N^{45\%} = 15$, $N^{60\%} = 45$. **D** $F_1$ score against data choice probability. If choices made up less than 50% of the data, the model had a poor predictive power. Dashed ellipses visualize clustering of the data with high and low $F_1$ score. **E** Comparison of generative properties of the three RL models: Return probability. **F** Comparison of generative properties of the three RL models: Return run lengths. Red curves: exponential fits.
(PDF)

## Acknowledgments

We gratefully acknowledge helpful comments from Dennis Eckmeier, Alex Gomez-Marin, Daisuke Hattori and Ollie Hulme.

## Author Contributions

**Conceptualization:** Duda Kvitsiani.

**Data curation:** Sophie E. Seidenbecher, Duda Kvitsiani.

**Funding acquisition:** Duda Kvitsiani.

**Investigation:** Duda Kvitsiani.

**Methodology:** Sophie E. Seidenbecher, Joshua I. Sanders.

**Resources:** Duda Kvitsiani.

**Software:** Sophie E. Seidenbecher, Joshua I. Sanders.

**Supervision:** Duda Kvitsiani.

**Validation:** Sophie E. Seidenbecher.

**Visualization:** Anne C. von Philipsborn.

**Writing – original draft:** Sophie E. Seidenbecher, Duda Kvitsiani.

**Writing – review & editing:** Joshua I. Sanders, Anne C. von Philipsborn, Duda Kvitsiani.

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
