## [Decision Letter · Decision Letter 0]

8 Jul 2020

PONE-D-20-14946

Reward foraging task and model-based analysis reveal how fruit flies learn value of available options

PLOS ONE

Dear Dr. Kvitsiani

Thank you for submitting your manuscript to PLOS ONE. After careful consideration, we feel that it has merit but does not fully meet PLOS ONE’s publication criteria as it currently stands. Therefore, we invite you to submit a revised version of the manuscript that addresses the points raised during the review process.

Although the manuscript was received very well, both reviewers recommend changes and request data availability. Please address all questions and recommendations as in my opinion will improve the manuscript substantially.

We look forward to receiving your revised manuscript.

Kind regards,

Efthimios M. C. Skoulakis, PhD

Academic Editor

PLOS ONE

Journal Requirements:

"No authors have competing interests"

We note that one or more of the authors are employed by a commercial company: Sanworks LLC.

2.1. Please provide an amended Funding Statement declaring this commercial affiliation, as well as a statement regarding the Role of Funders in your study. If the funding organization did not play a role in the study design, data collection and analysis, decision to publish, or preparation of the manuscript and only provided financial support in the form of authors' salaries and/or research materials, please review your statements relating to the author contributions, and ensure you have specifically and accurately indicated the role(s) that these authors had in your study. You can update author roles in the Author Contributions section of the online submission form.

2.2. Please also provide an updated Competing Interests Statement declaring this commercial affiliation along with any other relevant declarations relating to employment, consultancy, patents, products in development, or marketed products, etc. 

Reviewers' comments:

Reviewer's Responses to Questions

**Comments to the Author**

1. Is the manuscript technically sound, and do the data support the conclusions?

Reviewer #1: Yes

Reviewer #2: Yes

2. Has the statistical analysis been performed appropriately and rigorously? 

Reviewer #1: Yes

Reviewer #2: Yes

3. Have the authors made all data underlying the findings in their manuscript fully available?

Reviewer #1: Yes

Reviewer #2: No

4. Is the manuscript presented in an intelligible fashion and written in standard English?

Reviewer #1: Yes

Reviewer #2: Yes

5. Review Comments to the Author

Reviewer #1: Seidenbrecher et al investigate reward driven behaviour in single flies. In their assay, flies walk in linear arena and receive fictive reward - the optogenetic activation of sugar sensing gustatory receptor neurons – when entering one of two distal zones. The authors report that, similar to previous work, activating sugar sensing neurons lead to a change of behaviour and results in a preference for the rewarded location. By varying the stimulation protocol, the authors show that the probability of fictive reward positively correlates with the formation of the place preference. The bias to spend more time in the fictive-reward zone seems to arise from stimulation induced stops and turns and, maybe more interestingly, from a learned response: the return into the fictive reward zone in unrewarded trials. The authors suggest that the reward history influences the behaviour with the most recent reward being most influential. In contrast, Seidenbacher and colleagues claim that choice history does not correlate with future behaviour. Further, different reinforcement learning models are used to explain various phenomena of the observed behaviour, focusing on the update of value of unchosen options. Unfortunately, the later part lacks clarity in the description and the interpretation, which the authors should address before publication. In general, the presented paradigm is one of very view probabilistic learning assays in fruit flies and therefore will potentially pave the way for future studies on how accumulated value drives goal driven behaviour in the fly. However, the authors should consider addressing the following points before publication:

1) Seidenbacher and colleagues observe a decay of the place preference over time especially in high reward-probability settings. They interpret this observation as a desensitization and discuss it in the context of the lack of nutritional input. However, there is an alternative explanation for the phenomenon which the authors should consider: flies may learn that the artificially driven sensation of sugar is not followed by the presence of caloric food. Evidence for caloric frustration learning with sweet-only sugars has been previously reported in flies and therefore should be at least discussed (Musso et al 2017 Nat Com).

2) The impact of past choices on future behaviour seem to be limited, since the authors find no effects of past returns on future returns. However, it is not clear if the authors consider if a return leads to reward or to reward omission. After rewarded trials a non-rewarded return is conceptually similar to the omission of reward during memory extinction experiments in the fly (Felsenberg et al 2017). Reward memory extinction seems to drive the formation of an opposing memory. Thus, it might well be that unrewarded returns have a different impact on the following decisions compared to rewarded returns. The authors should be able to test for this hypothesis to strengthen their claim that choice history does not impact future choices.

3) The authors compare different learning models to “…see how unchosen option values are updated.”(line 235-266). This paragraph needs more clarity in stating its goal and the conclusion, e.g. the authors should elaborate on the differences between the chosen unrewarded and unchosen option trials, why it is important to understand how values are updated in unchosen option trials, and what it means that the behaviour is captured by the specific model. In addition, it would improve the manuscript if the authors could define forgetting, e.g. active vs passive forgetting (Davis and Zhong 2017 Neuron).

4) One substantial observation is that flies have an increased return probability early after being placed in the chambers, independent from the given reward. The authors argue that this particular bias in returns should be interpreted as an attraction to the edges of the arena and therefore include an initial positive bias into their model. However, there is no evidence that this increased return behaviour is driven by an appetitive response, e.g. the bias towards the edge of the arena could result from an escape behaviour. In such a setting the fly might be in an aversive state which could have consequences for the evaluation of an occurring reward. The authors should include such an alternative scenario into their interpretation.

Minor aspects:

-Figure 2a, I would encourage the authors to improve the visual display of what a trial is and show including the definition of a return decision is and what not.

-I would advise to turn down the claim that the assay is a value-based decision task. One could also call it a navigational task or simple place learning.

Reviewer #2: In this study, the authors developed a single-fly assay to examine the effect of optogenetic appetitive reinforcement on foraging in a simple behavioral arena. They used it to test how probabilistic reinforcement schedules affects behavioral responses in behaving flies, testing several major theoretical models that are proposed to underlie reinforcement learning. The data and figures are nicely presented, and I have no major experimental concerns. I have several comments about data availability and suggestion to improve the clarity of the presentation.

Major points

Data availability is not sufficient. Raw data should be uploaded to a repository to facilitate third-party access, which is particularly important for a manuscript such as this, where others may wish to pour over the models and reanalyze the findings.

The authors should also consider uploading scripts and code, as the apparatus would be beneficial to other researchers in the field, and broader implementation would increase the impact of the work.

Minor points

In figure 1D,G,H, the corresponding legend captions, and text line 111, the genetic controls need to be specified (Gal4/+ or UAS/+).

In figure 1F and 2B legend: “genetic controls” should be more precisely defined, so that they are understandable for non-Drosophila researchers.

Line 136: typo (affect vs. effect); should read: …we did not see any effect of …

6. PLOS authors have the option to publish the peer review history of their article (what does this mean?). If published, this will include your full peer review and any attached files.

Reviewer #1: No

Reviewer #2: No

---

## [Author Response · Author response to Decision Letter 0]

6 Aug 2020

Rebuttal Letter

Reviewers Comments to the Author

Reviewer #1: Seidenbrecher et al investigate reward driven behaviour in single flies. In their assay, flies walk in linear arena and receive fictive reward - the optogenetic activation of sugar sensing gustatory receptor neurons – when entering one of two distal zones. The authors report that, similar to previous work, activating sugar sensing neurons lead to a change of behaviour and results in a preference for the rewarded location. By varying the stimulation protocol, the authors show that the probability of fictive reward positively correlates with the formation of the place preference. The bias to spend more time in the fictive-reward zone seems to arise from stimulation induced stops and turns and, maybe more interestingly, from a learned response: the return into the fictive reward zone in unrewarded trials. The authors suggest that the reward history influences the behaviour with the most recent reward being most influential. In contrast, Seidenbacher and colleagues claim that choice history does not correlate with future behaviour. Further, different reinforcement learning models are used to explain various phenomena of the observed behaviour, focusing on the update of value of unchosen options. Unfortunately, the later part lacks clarity in the description and the interpretation, which the authors should address before publication. In general, the presented paradigm is one of very view probabilistic learning assays in fruit flies and therefore will potentially pave the way for future studies on how accumulated value drives goal driven behaviour in the fly. However, the authors should consider addressing the following points before publication:

1) Seidenbacher and colleagues observe a decay of the place preference over time especially in high reward-probability settings. They interpret this observation as a desensitization and discuss it in the context of the lack of nutritional input. However, there is an alternative explanation for the phenomenon which the authors should consider: flies may learn that the artificially driven sensation of sugar is not followed by the presence of caloric food. Evidence for caloric frustration learning with sweet-only sugars has been previously reported in flies and therefore should be at least discussed (Musso et al 2017 Nat Com).

2) The impact of past choices on future behaviour seem to be limited, since the authors find no effects of past returns on future returns. However, it is not clear if the authors consider if a return leads to reward or to reward omission. After rewarded trials a non-rewarded return is conceptually similar to the omission of reward during memory extinction experiments in the fly (Felsenberg et al 2017). Reward memory extinction seems to drive the formation of an opposing memory. Thus, it might well be that unrewarded returns have a different impact on the following decisions compared to rewarded returns. The authors should be able to test for this hypothesis to strengthen their claim that choice history does not impact future choices.

3) The authors compare different learning models to “…see how unchosen option values are updated.”(line 235-266). This paragraph needs more clarity in stating its goal and the conclusion, e.g. the authors should elaborate on the differences between the chosen unrewarded and unchosen option trials, why it is important to understand how values are updated in unchosen option trials, and what it means that the behaviour is captured by the specific model. In addition, it would improve the manuscript if the authors could define forgetting, e.g. active vs passive forgetting (Davis and Zhong 2017 Neuron).

4) One substantial observation is that flies have an increased return probability early after being placed in the chambers, independent from the given reward. The authors argue that this particular bias in returns should be interpreted as an attraction to the edges of the arena and therefore include an initial positive bias into their model. However, there is no evidence that this increased return behaviour is driven by an appetitive response, e.g. the bias towards the edge of the arena could result from an escape behaviour. In such a setting the fly might be in an aversive state which could have consequences for the evaluation of an occurring reward. The authors should include such an alternative scenario into their interpretation.

Minor aspects:

-Figure 2a, I would encourage the authors to improve the visual display of what a trial is and show including the definition of a return decision is and what not.

-I would advise to turn down the claim that the assay is a value-based decision task. One could also call it a navigational task or simple place learning.

Reviewer #2: In this study, the authors developed a single-fly assay to examine the effect of optogenetic appetitive reinforcement on foraging in a simple behavioral arena. They used it to test how probabilistic reinforcement schedules affects behavioral responses in behaving flies, testing several major theoretical models that are proposed to underlie reinforcement learning. The data and figures are nicely presented, and I have no major experimental concerns. I have several comments about data availability and suggestion to improve the clarity of the presentation.

Major points

Data availability is not sufficient. Raw data should be uploaded to a repository to facilitate third-party access, which is particularly important for a manuscript such as this, where others may wish to pour over the models and reanalyze the findings.

The authors should also consider uploading scripts and code, as the apparatus would be beneficial to other researchers in the field, and broader implementation would increase the impact of the work.

Minor points

In figure 1D,G,H, the corresponding legend captions, and text line 111, the genetic controls need to be specified (Gal4/+ or UAS/+).

In figure 1F and 2B legend: “genetic controls” should be more precisely defined, so that they are understandable for non-Drosophila researchers.

Line 136: typo (affect vs. effect); should read: …we did not see any effect of …

Our response

We thank reviewers for critical comments and helpful suggestions. We address in our revised manuscript all the points raised by reviewers.

 Reviewer 1.

1. We removed the term desensitization from our description of the behavioral responses as it may be misleading to characterize the flies decisions as merely sensory driven processes. We now consider in our results (line X-Y) as well as in our discussion section (line 200-220, 228,402, 404,) the possibility that the reduction in flies returns to rewarded location could be due to Caloric frustration memory. 

2. We apologize for being not very clear on this topic. When we analyze the return behavior using regression analysis, where we regress current returns against both previous rewards and previous choices. In Materials and Methods we now correctly describe the regression model (line 507-510, equation 2). Thus, if only past rewards contributed to future returns , past returns should have had no effect. This is what we saw and that made us conclude that past returns had no effect. 

3. We apologizer again for not being clear. There was no need to distinguish unrewarded and unchosen options. We simply meant to distinguish the value update process on unchosen vs chosen options (line 250). We corrected it. Furthermore, we updated the same section by introducing the concept of forgetting and how it may relate to foraging in our assay (line 251-256). We also explain how different models were compared (line 260-272) and finally include our conclusion for that section (line 295-296). In our discussion part we again discuss the role of active forgetting and why we think this is what we see in our assay (line 442-448). We particularly thank reviewer 1 for raising this point as we think it places our results in a wider context of fruit fly learning and memory field.

4. In the updated version, we are discussing the possibility that the high return rate at the beginning of the session could be due to the aversive state of flies. It is very much conceivable that when placed in a new and enclosed environment flies become motivated to escape, which may result in high incidence of turning behavior. We think this scenario is unlikely as it would be difficult to explain why first rewards in aversive state should also result in high return rates (lines 349-355).

Minor:

We also turned down the claim that this is a value-based decision making tasks and rephrased it as the reward foraging task. (Lines30, 350, 379,384)

We also updated the figure 2A, expanded it and indicated the return and non-returns in figure2 legends 

Reviewer 2.

Major points

We deposited the fly tracking data on figshare.com and it will be freely available upon publication of the MS. We also deposited the code that runs the closed loop tracking and optical stimulation protocol. It is a custom written code in Matlab and C++ environment that tracks 12 flies separately and triggers short pulses when certain conditions are met, i.e. when the fly enters the trigger and reward zones. 

The analysis code for behavior will be available on request. 

Minor points

1. We incorporate designation of the genetic controls in the figure legends in figure1 and figure2.

2. We corrected the typo.

---

## [Decision Letter · Decision Letter 1]

3 Sep 2020

PONE-D-20-14946R1

Reward foraging task and model-based analysis reveal how fruit flies learn value of available options

PLOS ONE

Dear Dr. Kvitsiani,

Thank you for submitting your manuscript to PLOS ONE. After careful consideration, we feel that it has merit but does not fully meet PLOS ONE’s publication criteria as it currently stands. Therefore, we invite you to submit a revised version of the manuscript that addresses the points raised during the review process.

First of all sorry for the delay as one of the reviewers was indisposed as i understand. You only need to make the minor changes suggested by reviewer 2 and the manuscript will be good to go

We look forward to receiving your revised manuscript.

Kind regards,

Efthimios M. C. Skoulakis, PhD

Academic Editor

PLOS ONE

Reviewers' comments:

Reviewer's Responses to Questions

**Comments to the Author**

1. If the authors have adequately addressed your comments raised in a previous round of review and you feel that this manuscript is now acceptable for publication, you may indicate that here to bypass the “Comments to the Author” section, enter your conflict of interest statement in the “Confidential to Editor” section, and submit your "Accept" recommendation.

Reviewer #1: All comments have been addressed

Reviewer #2: (No Response)

2. Is the manuscript technically sound, and do the data support the conclusions?

Reviewer #1: Yes

Reviewer #2: Yes

3. Has the statistical analysis been performed appropriately and rigorously? 

Reviewer #1: Yes

Reviewer #2: Yes

4. Have the authors made all data underlying the findings in their manuscript fully available?

Reviewer #1: Yes

Reviewer #2: Yes

5. Is the manuscript presented in an intelligible fashion and written in standard English?

Reviewer #1: Yes

Reviewer #2: Yes

6. Review Comments to the Author

Reviewer #1: The authors have addressed my concerns and improved on the clarity of the manuscript. I recommend the article for publication in the current form and I am looking forward to the future usage of the assay and the frame work.

Reviewer #2: Fig. 1 D,G, and H still need labels to specify control genotypes. Other than that, the authors have addressed all of my concerns.

7. PLOS authors have the option to publish the peer review history of their article (what does this mean?). If published, this will include your full peer review and any attached files.

Reviewer #1: No

Reviewer #2: No

---

## [Author Response · Author response to Decision Letter 1]

9 Sep 2020

Reviewers Comments to the Author

Reviewer #2: Fig. 1 D,G, and H still need labels to specify control genotypes. Other than that, the authors have addressed all of my concerns. 

Our response

We addressed this issue and indicated clearly the control genotypes for Fig. 1D, G, and H. At the beginning of the Fig.1 legend, the second sentence, we also say what do we mean by gen.controls.

---

## [Editor Report · Decision Letter 2]

10 Sep 2020

Reward foraging task and model-based analysis reveal how fruit flies learn value of available options

PONE-D-20-14946R2

Dear Dr. Kvitsiani,

We’re pleased to inform you that your manuscript has been judged scientifically suitable for publication and will be formally accepted for publication once it meets all outstanding technical requirements.

Kind regards,

Efthimios M. C. Skoulakis, PhD

Academic Editor

PLOS ONE
---

## [Editor Report · Acceptance letter]

22 Sep 2020

PONE-D-20-14946R2 

Reward foraging task and model-based analysis reveal how fruit flies learn value of available options 

Dear Dr. Kvitsiani:

I'm pleased to inform you that your manuscript has been deemed suitable for publication in PLOS ONE. Congratulations! Your manuscript is now with our production department. 

Kind regards, 

on behalf of

Dr. Efthimios M. C. Skoulakis 

Academic Editor

PLOS ONE